# Persistent Müllerian Duct Syndrome with Supernumerary Testicles Due to a Novel Homozygous Variant in the *AMHR2* Gene and Literature Review

**DOI:** 10.3390/diagnostics14232621

**Published:** 2024-11-21

**Authors:** Luminita Nicoleta Cima, Iustina Grosu, Isabela Magdalena Draghici, Augustina Cornelia Enculescu, Adela Chirita-Emandi, Nicoleta Andreescu, Maria Puiu, Carmen Gabriela Barbu, Simona Fica

**Affiliations:** 1Endocrinology Department, Elias Emergency University Hospital, 011461 Bucharest, Romania; luminita.cima@umfcd.ro (L.N.C.); iustina.grosu@rez.umfcd.ro (I.G.); carmen_gabriela_barbu@yahoo.co.uk (C.G.B.); simonafica55@gmail.com (S.F.); 2Faculty of General Medicine, Carol Davila University of Medicine and Pharmacy, 050474 Bucharest, Romania; 3Pediatric Surgery Department, Maria Sklodowska Curie Emergency Hospital for Children, 077120 Bucharest, Romania; 4Pathology Department, Maria Sklodowska Curie Emergency Hospital for Children, 077120 Bucharest, Romania; augustina.enculescu@gmail.com; 5Department of Microscopic Morphology, Genetics Discipline, Center of Genomic Medicine, University of Medicine and Pharmacy “Victor Babeș”, 400347 Timișoara, Romania; adela.chirita@umft.ro (A.C.-E.); andreescu.nicoleta@umft.ro (N.A.); maria_puiu@umft.ro (M.P.); 6Regional Center of Medical Genetics Timiș, Clinical Emergency Hospital for Children “Louis Țurcanu”, 300011 Timișoara, Romania

**Keywords:** persistent Müllerian duct syndrome, supernumerary testes, congenital anomalies, genetic testing, *AMHR2* gene

## Abstract

**Introduction**: Persistent Müllerian duct syndrome (PMDS) is a rare disorder of sex development (DSD) caused by mutations in the genes coding anti-Müllerian hormone (AMH) or the AMH receptor, characterized by the persistence of Müllerian derivatives, the uterus and/or fallopian tubes, in otherwise normally virilized boys. Testicular regression syndrome is common in PMDS, yet the association with supernumerary testis has been reported in only two patients where genetic testing was not performed. **Method**: Thus, we report an individual with this particular association caused by a previously unreported homozygous variant in the *AMHR2* gene to enable future genotype–phenotype correlations in this rare disorder. In addition, a search of PMDS associated with congenital anomalies reported in the literature was performed to provide a comprehensive overview of this pathology. **Results**: We present the case of a 13-year-old boy with a history of bilateral cryptorchidism. Two attempts of right orchidopexy were performed at the age of 4 and 5 years. At that time, exploratory laparoscopy identified an intra-abdominal left testicle. In addition, a fibrous structure extending from the left intra-abdominal testicle to the deep inguinal ring (Müllerian duct remnants) and a medially located abdominal mass, bilaterally fixated to the parietal peritoneum (uterine remnant), were detected. The left testicular biopsy revealed immature prepubertal testicular tissue. The uterine remnant was dissected and removed and the left orchidopexy was performed. The karyotype was 46, XY without other numerical or structural chromosomal abnormalities. Reinterventions on the left testicle were performed at the age of 9 and 12 years when a testicular remnant was identified in the left inguinal canal and removed. Three months after left orchidectomy, ultrasound followed by abdominopelvic MRI identified a structure resembling a testis in the left inguinal area. Another surgical exploration was performed, and a mass located outside (lateral) the inguinal canal was found. A biopsy from the suspected mass was performed. The histopathologic examination showed characteristics of immature prepubertal testis. The patient was later referred to our clinic with the suspicion of DSD. Serum AMH and inhibin B were normal. Therefore, the diagnosis of PMDS was suspected. Genetic testing was performed using next-generation sequencing in a gene panel that included *AMH* and *AMHR2* genes. A homozygous variant classified as likely pathogenic in the *AMHR2* gene was identified but remains unreported in the literature (NC_000012.11:g.53823315T>C in exon 8 of the *AMHR2* gene). **Conclusions**: A high degree of suspicion and awareness is needed to diagnose this condition in order to avoid iterative surgery. The coexistence of two extremely rare conditions (PMDS and supernumerary testes) has been reported previously in only two patients, yet the association could have a common pathophysiologic background. Our case, reporting a novel AMHR2 variant, highlights the importance of genetic testing in these individuals in order to elucidate a possible genotype–phenotype correlation.

## 1. Introduction

Persistent Müllerian duct syndrome (PMDS) is a rare disorder of sex development (DSD) caused by mutations in the genes coding anti-Müllerian hormone (AMH) or the AMH receptor, characterized by the persistence of Müllerian derivatives (MDs), the uterus and/or fallopian tubes, in otherwise normally virilized boys [1]. At the time of writing, fewer than 300 cases have been reported in the literature. Testicular regression syndrome is common in PMDS, yet the association with supernumerary testis has been reported in only two individuals, in whom genetic testing was not performed [2,3]. Thus, we aim to report an individual with this particular association caused by a previously unreported homozygous variant in the *AMHR2* gene to enable future genotype–phenotype correlations in this rare disorder. In addition, a search of PMDS associated with congenital anomalies reported in the literature was performed to provide a comprehensive overview of this pathology.

## 2. Case Report

We present the case of a patient diagnosed with PMDS during laparoscopic exploration for bilateral cryptorchidism, who was later diagnosed with polyorchidism in the context of redo interventions for recurrent undescended testes. This study was conducted in compliance with the Declaration of Helsinki (updated in 2013). Informed written approval was acquired from the patient (guardian of the patient) for the publication of this case report and the associated photos.

### 2.1. At Age 4 Years

A full-term male, without relevant family or prenatal history, was evaluated at 4 years of age for bilateral cryptorchidism and bilateral inguinal hernia. The patient had no history of parents’ consanguinity. On clinical examination, the external genital organs had a male phenotype, with a stretched penile length within age-appropriate limits and a normally located external urethral meatus. At this point, no hormonal tests were performed in order to assess gonadal function. Only TSH, FT4, and prolactin were collected with normal values. Pelvic MRI revealed an intrabdominal, hypoplastic right testis and another structure, situated in the left paramedian retrovesical area, resembling an atrophic testis. At that time (February 2013), the patient underwent bilateral inguinotomy under general anesthesia. Right orchidopexy with pelvic fixation, exploration of the left inguinal region, and bilateral herniorrhaphy were performed. During the exploration of the left inguinal region, the left testicle was not detected; thus, the left orchidopexy could not be performed at that time.

### 2.2. At Age 5 Years

After orchidopexy, the right testis ascended again in the abdomen, and in May 2014, at the age of 5, the patient presented for the first time to our clinic (“Maria Sklodowska Curie” Emergency Hospital for Children, Bucharest, Romania). The preoperative investigation (MRI) revealed the presence of a left paramedian retrovesical lesion, possibly the left ectopic testicle, as well as a right testicle located in the upper 1/3 of the right inguinal canal. The Pregnyl stimulation test indicated viable testicular tissue. Under general anesthesia, exploratory laparoscopy was performed, which revealed a left intra-abdominal testicle with a modified appearance and an anatomical fibrous structure medially located that was bilaterally fixated to the parietal peritoneum, extended from the left testis to the right deep inguinal ring, resembling a rudimentary uterus. A biopsy of the left ectopic testicle was performed and the histopathologic exam showed characteristics of immature testicular tissue (Figure 1). At this point, an iterative right inguinotomy was also performed, which detected in the right inguinal canal a testicle fixed at this level (probably during the first surgical intervention, which was not performed in our clinic), and a right orchidopexy was carried out. Laparoscopic exploration was repeated in October 2014, during which the structure was dissected and removed and left orchidopexy was performed. Histopathology examination confirmed the Müllerian nature of the mass (Figure 2). The karyotype was 46, XY and the sex-determining region (SRY) on the Y chromosome was identified via fluorescence in situ hybridization (FISH).

### 2.3. At Age 9–12 Years

The follow-up of the patient consisted of annual scrotal ultrasound examinations. In April 2018, at the age of 9, we observed that the left testicle had an elevated position, with it being at the level of the external opening of the left inguinal canal, and surgery was performed again through an incision at the level of the left hemiscrotum and the testicle was descended. On this occasion, we observed that the left testicle was hypotrophic. The right testicle was located in the right hemiscrotum.

In September 2021, at the age of 12, surgery was repeated on the left hemiscrotum, and a small atrophic left testicle was discovered, for which an orchidectomy was performed, taking into account the risk of malignancy. The histopathologic examination was consistent with epididymis, ductus deferens, without specific features of masculine gonads, and seminiferous tubules, respectively (Figure 3), which were initially seen at the biopsy performed in May 2014 from the left undescended testis (Figure 1). The left testis that was initially intra-abdominal and was descended and fixed into the scrotum at the age of 5 decreased in size in evolution (as it was discovered at the age of 9 when an iterative orchidopexy was performed) and finally crumbled (at the age of 12, the tissue that was removed had no gonadal tissue). The right testicle was still in the right scrotal bursa and measured 13/9/6 mm.

In February 2022, at the age of 13, following a routine ultrasound, a nodular image was discovered in the left inguinal region but outside the left inguinal canal, approximately 10–15 mm deep from the skin plane. The structure measured 15/10 mm and resembled a testis. The possibility of supernumerary gonads was raised. Another surgical intervention was performed and a biopsy from the suspected mass was retrieved. The histopathologic examination showed characteristics of immature prepubertal testis (Figure 4).

### 2.4. At Age 13 Years

At the age of 13 years, the patient was referred to our clinic with the suspicion of DSD. On physical examination, the patient’s height was 150.6 cm (−1.23 SD), their weight was 48.5 kg, their BMI = 21.4 kg/m^2^ (percentile 81), their Tanner stage was 2, and a small right testis was palpable in the scrotum, while the left one was identified in the inguinal area. Scrotal ultrasound was consistent with the clinical findings, showing a right hypoplastic right testis of 14/6.2 mm and a left testis of 16/8.9 mm in the inferior region of the inguinal area. Hormonal studies were performed to determine testicular function and revealed age-appropriate levels of luteinizing hormone (LH), follicle-stimulating hormone (FSH), and testosterone as follows: FSH = 3.03 mIU/mL (0.4–4.6), LH = 0.434 mIU/mL (0.1–7.8), testosterone = 3.12 ng/dL (Tanner 2: 2.5–432). Serum AMH and inhibin B were normal for age (serum AMH level 13.970 ng/mL; normal range: 2.079–30.656, serum inhibin B = 33.3 pg/mL; normal range: 16.610–278.870). Tumoral markers were within limits (LDH = 148 U/L, normal range: 105–233, AFP ≤ 0.908 ng/mL, normal range: <40, BETA-HCG ≤ 0.70 mIU/mL, normal range: <5).

Considering the male karyotype and the coexistence of testes, Wolffian duct structures, and Müllerian duct derivates, the diagnosis of PMDS was raised. Taking into account the normal levels of AMH, the *AMHR2* gene was believed to be causative.

### 2.5. Genetic Testing

Genetic testing was performed using next-generation sequencing (NGS) with the TruSightOne panel (Illumina, San Diego, CA, USA) on a MiSeq (Illumina) Sequencer at the Timis Regional Centre of Medical Genetics, affiliated with “Louis Turcanu” Emergency Hospital for Children in collaboration with the Center for Genomic Medicine Victor Babes University of Medicine and Pharmacy Timisoara. Libraries were generated according to the manufacturer’s protocols using TruSightOne kits (Illumina Inc., San Diego, CA, USA). Targeted DNA sequencing was performed on TruSightOne library v1.0, targeting 4813 genes. The secondary analysis used the Illumina MiSeq Reporter 2.6.2.3 platform, incorporating FASTQ alignment (using Burrows–Wheeler Align) and variant extraction (using SAMtools and GATK). Sequences were mapped to GRCh37 (“hg19”), retaining reads with a median quality score genotype quality (GQ) greater than 30, variant frequency greater than 20%, variant depth greater than 20, and strand bias less than −10. The VFC annotation was performed using ANNOVAR. The variant frequency datasets were analyzed using gnomAD version 4.1. In silico prediction relied on Combined Annotation-Dependent Depletion (CADD) scores as a tool that integrates multiple annotations such as conversion metrics, functional genomic data, transcript information, and protein level scores and computes a score that indicates the variant effect. All DNA sequencing results were manually reviewed by two clinical geneticists to prioritize variants and subsequent reporting of consensus variants. The sequencing analysis, bioinformatics filtering strategy, data interpretation, and reporting are also presented elsewhere [4]. Variants were classified according to the American College of Medical Genetics and Genomics (ACMG) guidelines in relation to the patient’s phenotype [5]. Pathogenic, likely pathogenic, and variants of uncertain significance (VUS) related to the phenotype were reported for clinical use. The test showed a likely pathogenic variant in a homozygous state, in *AMHR2* gene NM_020547.3:c.1046T>C, NP_065434.1:p.(Ile349Thr). This sequence change results in the replacement of isoleucine with threonine at position 349 of the AMHR2 protein (p.Ile349Thr) (Figure 5 showing the variant in Integrative Genomics Viewer (IGV)). Isoleucine 349 is located in a conserved region of the Protein Kinase functional domain, within AMHR2. UniProt protein AMHR2_HUMAN homo_sapiens proteome sequences have 65 missense/in-frame variants (22 pathogenic variants, 34 uncertain variants, and 9 benign variants), which qualifies as likely pathogenic. The detected variant is present in population databases with a very low frequency (gnomAD v2.1.1, ƒ = 0.00001591, no individual with homozygous status). To date, the variant has not been reported in other individuals with diseases associated with the *AMHR2* gene. Most of the in silico prediction algorithms used suggest that the variant could have a deleterious impact on the protein. In addition, the patient’s phenotype is suggestive and specific for disorders associated with the *AMHR2* gene. For these reasons, the variant was classified with likely pathogenic significance according to ACMG guidelines [5]. Unfortunately, carrier testing of the variants in the parents was not performed.

## 3. Literature Review

We conducted a comprehensive literature review on PMDS to provide an in-depth understanding of this rare pathology (Table 1). Our primary source of data was Medline-indexed studies accessed via PubMed. The aim was to include articles that reported cases of PMDS and highlighted clinical presentations, laboratory investigations, genetic and histopathological findings, and the outcomes of the disease, with a focus on the ones that reported the association with congenital anomalies. Our search included terms such as ‘Persistent Müllerian duct syndrome’ + ‘congenital anomalies’, ‘supernumerary testes’, ‘polyorchidism’, ‘genetic testing’, and ‘*AMHR2* gene mutations’ (Figure 6).

Following this protocol, we initially identified 301 articles, spanning from the earliest available date to 1 February 2024. After title and abstract screening, 228 eligible studies were sought for retrieval. Records were excluded from the research if they involved non-human studies, non-systematic reviews, other DSDs apart from PMDS, or abstracts in other languages. The full text of 19 reports could not be retrieved. In total, 14 additional relevant articles were also identified through a manual review of reference lists. The full text of the studies was reassessed and, based on a detailed evaluation of content, 61 articles mentioning an association between PMDS and at least one congenital malformation were ultimately included in the final manuscript. The reports that referred to the association between PMDS and hypospadias were removed from the analysis considering the fact that undervirilization is an indicator of testicular dysgenesis rather than PMDS.

## 4. Discussion

PMDS is a rare form of 46, XY DSD that presents a wide range of clinical manifestations, diagnostic findings, and treatment approaches. This discussion aims to compare the findings from a recent case with those from a comprehensive review of the cases published in the literature.

PMDS is a 46, XY DSD, characterized by an ineffective AMH, described by Nilson in 1939 [58]. In the absence of AMH action, Müllerian ducts persist and differentiate into rudimentary forms of the fallopian tubes, the uterus, and the upper third of the vagina (due to the absence of estrogen). Caudally, these structures open into the posterior aspect of the urethra, near the verumontanum [59]. Leydig cells are functional, so virilization is complete, and Wolffian duct derivates are preserved [1]. The vas deferens pass nearby or are encapsulated within the uterine walls and open into the upper vagina, the female equivalent of the prostatic utricle. Excretory duct anomalies are nevertheless frequent in PMDS [60].

Our recent case refers to a patient without relevant family or prenatal history who was first evaluated at the age of 4 years for bilateral cryptorchidism and bilateral inguinal hernia. The clinical presentation of our patient is consistent with the female form of PMDS which occurs in 80% of cases [61]. The female type presents with bilateral impalpable testis (the testes remain intra-abdominal in a high ovarian position) and is usually diagnosed during laparoscopic exploration [62]. In a minority of cases, testis may prolapse, due to the long, flexible gubernaculum, into the processus vaginalis, dragging along the MD (male form). Classical male presentations are hernia uteri inguinalis (a hernia sac accommodating the testis and the MD derivates) or transverse testicular ectopia (an inguinal hernia containing both testes, the uterus, and the fallopian tubes) [60,63]. Usually, the male types (20%) are discovered during herniorrhaphy.

AMH is involved in the swelling reaction of the genito-inguinal ligament. In PMDS, this process is altered and the gubernaculum remains elongated, similar to the round ligament. Consequently, testes may reascend after orchidopexy, and this explains the necessity of recurrent bilateral orchidopexies in our patient.

The coexistence of Müllerian and Wolffian duct remnants can be encountered in other forms of DSD apart from PMDS. Such examples are partial testicular dysgenesis or ovotesticular DSD [64,65,66,67]. In contrast with PMDS, these conditions generally present with ambiguous genitalia and asymmetrically formed internal genital organs [1]. If there is any uncertainty, gonadal biopsies and a karyotype should be performed [63]. In our case, complete external virilization pointed toward PMDS. Nevertheless, considering that 10% of patients with ovotesticular DSDs present with a normal or almost normal penis [68] and taking into account the atypical clinical presentation (abnormal appearance and supernumerary gonads), further investigations were conducted. The 46, XY karyotype and the presence of normal testicular tissue support the working diagnosis.

Hormonal studies revealed that normal testicular function and serum AMH and inhibin B were normal for age in the present case. Testosterone production is usually unaffected in PMDS patients. Seric AMH levels can suggest the pathophysiology of the condition. Type 1 is routinely associated with low AMH values, while in type 2, AMH levels are normal or high [1,69].

PMDS shows an autosomal recessive inheritance [70], but de novo mutations are also possible [60,63,71,72]. Homozygous or compound heterozygous alterations in *AMH* associated with loss of function (type 1) or *AMHR2* determining receptor resistance (type 2) [73] have been identified in approximately 88% of PMDS cases [60,74]. In a minority of cases, no specific variant has yet to be identified (idiopathic PMDS). The detection of PPP1R12A truncation mutations coding myosin phosphatase in five cases of PMDS suggests that myosin phosphatase is involved in Müllerian regression, independently of the AMH signaling cascade [49]. It seems that myosin phosphatase is required for cell mobility, which plays a major role in Müllerian regression; alternatively, PPP1R12A mutations could affect the AMH transduction pathway [49]. In our patient, a homozygous variant classified as likely pathogenic in the *AMHR2* gene was identified through NGS, yet it remains unreported in the literature.

It has been reported that the *AMHR2* gene is located on the long arm of chromosome 12 and that it contains eleven exons: three exons form the extracellular domain, responsible for binding AMH specifically, the fourth encodes the transmembrane region, and the remaining seven account for the intracellular domain with serine/threonine kinase function [70,73]. Natural mutations of the AMH type II receptor found in PMDS affect ligand binding, signal transduction, and cellular transport [75,76]. The missense variant described in our patient determines the replacement of the 349th amino acid, a highly conserved region in exon 8, from isoleucine to threonine (p.Ile349Thr), altering protein kinase’s function. However, functional studies have not been conducted; therefore, further work is needed to investigate the harmfulness of this variant.

Up to now, the most common genetic test performed to detect mutations of *AMH* or *AMHR2* in relation to PMDS is Sanger sequencing with a few exceptions [8,74,77]. However, as suggested by Tosca et al., in the near future, NGS targeted to *AMH* and *AMHR2* may become the method of choice for diagnosing PMDS, with the capacity for detecting both SNVs and CNVs in a single step [8]. In our patient, the variant was identified through NGS and, to the best of our knowledge, it has not been reported before in the literature.

Of particular interest in our case is the association between PMDS and supernumerary testes in the presence of a genetic diagnosis of this rare disease. Usually, congenital malformations, notably of intestinal and renal nature, are described in PMDS patients. In most cases, no significant variants in *AMH* or *AMHR2* genes have been identified (idiopathic PMDS) [24,49,60]. Testicular abnormalities have also been reported in PMDS patients, mainly testicular regression syndrome. Testicular torsion is most probably the cause, due to the increased mobility of gonads and the long gubernaculum [78]. At the other end of the spectrum, the association with polyorchidism that we discovered in our patient is less often encountered. Polyorchidism is defined by the presence of more than two testes. At the present time, around 200 cases have been reported. The cause is still not established, but one explanation may be the division of the undifferentiated gonadal ridge by congenital peritoneal bands [79]. Apart from the presented case, only two other cases of PMDS with supernumerary testes have been documented. The first one presents a 46, XY patient with left hernia uteri inguinalis, left intra-abdominal testis, and two right scrotal testes [2]. The second case describes the occurrence of malignant degeneration of a supernumerary testis in a 46, XY patient with a female form of PMDS [3]. In both cases, no further genetic testing was performed. However, we would like to acknowledge the limitations regarding the evidence of a supernumerary testis in our case. At the age of 12 years, the tissue that was removed from the left side did not have gonadal tissue, although seminiferous tubules were initially seen at the biopsy performed in May 2014 (at the age of 5) from the left undescended testis. We excluded a testicular–epididymal dissociation on the left side because, in this case, the testicle remained fixed in the scrotum, while the vas deferens and the epididymis ascended into the inguinal canal. In our case, the epididymis was found in the scrotum (at the age of 12) and the testicle was found outside (anterolateral to) the inguinal canal (at the age of 13). Apart from this, the left testicle (which was initially intra-abdominal and was descended and fixed into the scrotum at the age of 5) was smaller than the right testicle, was of only a few mm, and decreased in size progressively (observed at the age of 9), until it crumbled (at the age of 12). The testicle found outside the left inguinal canal at the age of 13 was bigger than 1.5 cm; thus, it cannot be the same structure descended and fixed into the left scrotum. Therefore, we consider this to be strong evidence that the lesion identified at the age of 13 is a supernumerary testis.

As PMDS patients are at risk of malignant testicular degeneration, a differential diagnosis between a supernumerary testis and a lymph node metastasis from a testicular or MD remnant cancer was considered. Tumoral markers were within limits in our patient and the biopsy showed characteristics of immature testis. It has been shown that testicular malignant degeneration occurs in 33% of adults with PMDS [80]. A recent search on Medline/PubMed retrieved 44 articles (49 patients) on testicular tumors associated with PMDS, with the majority (59%) presenting with a large abdominal mass [81], but only five cases (10%) had a preceding history of appropriately managed cryptorchidism. The most commonly reported tumors in the literature were seminomas [82,83,84,85,86,87], but other types of tumors such as mixed germ cell tumors [88,89,90,91,92], embryonal carcinoma, yolk sac tumors, and teratoma were also encountered [63,93,94,95]. Some studies cite an incidence similar to that of cryptorchidism (5–18%) [63], while others suggest that the real incidence may be as high as 33% [60]. The American Urologic Association recommends early orchidopexy in patients with undescended testis by 6 months to decrease the risk of malignancy and emphasizes the importance of testicular self-examination after puberty for early detection [96]. At the time of writing, 12 cases of uterine neoplasma have been reported, affecting patients as young as 4 years [63]. Adenocarcinoma, adenosarcoma, and squamous cell carcinoma are the histopathological types found, and some case reports describe highly aggressive tumors [38,43,97,98]. Removal of Müllerian structures eliminates the risk. If resection is impossible out of fear of damaging the gonads or the excretory ducts; close surveillance consisting of regular pelvic ultrasounds or MRI should be envisioned [23,99]. Occasionally, other types of cancers have been described in patients with PMDS such as prostatic adenocarcinoma [100,101].

Even though clear guidelines for the management of PMDS have not yet been established, the main objectives that experts agree upon are the preservation of fertility and the prevention of malignant degeneration [1,60,63,102]. Unfortunately, despite a few reports of PMDS patients being able to father children, either spontaneously [83,84,103] or through assisted reproduction treatments [19,60,104], most affected males are infertile [1,105]. The mechanisms involved are cryptorchidism and late orchidopexy, male excretory duct anomalies, and iatrogenic lesions. In order to minimize this risk, early correction of cryptorchidism should be performed. In our case, the orchidopexies were performed after the age of 4 when the patient first approached the medical system. Müllerian structures pose a mechanical constraint to testicular descent and are therefore usually resected. During dissection, great attention should be paid to the conservation of the vas deferens and blood vessels, as these structures are in close proximity or even enclosed within the uterine walls [60]. As the vas deferens and vascular structures may be injured during orchiopexy, leading to subsequent infertility, patients should always be referred to experienced clinics [60]. If spermatogenesis is present, testicular sperm extraction followed by ICSI may be used to facilitate conception in cases of congenital or iatrogenic lesions of the male excretory ducts [8].

In our case, a dissection of the hypoplastic uterus was performed before the diagnosis of PMDS. Treatment of the remnants of MD remains controversial [102,106]. Previous studies have demonstrated that children with PMDS require the removal of MD remnants as they are also prone to malignant transformation [25,107] and predispose to urinary tract infections, periodic hematuria, stones, and urination disorders because they are connected with the seminal vesicle [108,109]. As PMDS patients have been reported to develop malignancy of Müllerian remnants originating from the mucosa, some authors suggest that the preferred surgical procedure should split the uterus in the middle, destroy the mucosal lining, and leave an intact pedicle of the myometrium. This procedure releases the testes to the ideal position and also protects the integrity and vascularity of the vas deferens and reduces the chance of malignancy [6,51,110].

## 5. Conclusions

A high degree of suspicion and awareness is needed to diagnose this condition in order to avoid iterative surgery. The coexistence of two extremely rare conditions (PMDS and supernumerary testes) has previously been reported in only two patients, yet the association could have a common pathophysiologic background. Our case, reporting a novel AMHR2 variant, highlights the importance of genetic testing in these individuals in order to elucidate a possible genotype–phenotype correlation. Further genetic studies on these individuals are needed in order to investigate this unusual association.

## Figures and Tables

**Figure 1 diagnostics-14-02621-f001:**
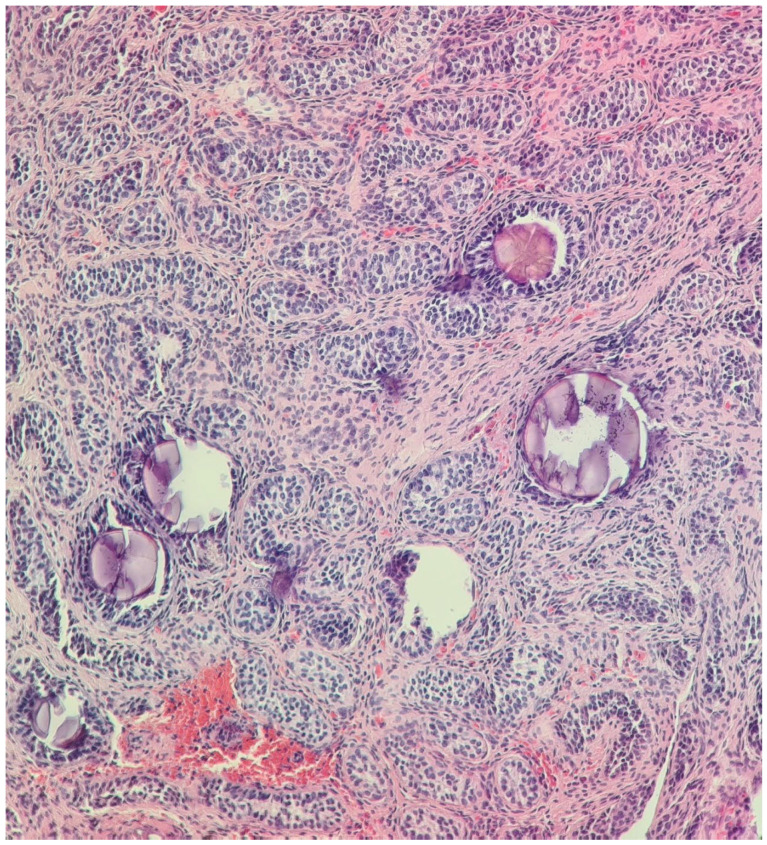
Undescended left testis biopsy: immature testis consisting of small seminiferous tubules without lumen and composed of immature Sertoli cells, with no evidence of spermatogenesis (May 2014). H&E stain, 200× and 400×.

**Figure 2 diagnostics-14-02621-f002:**
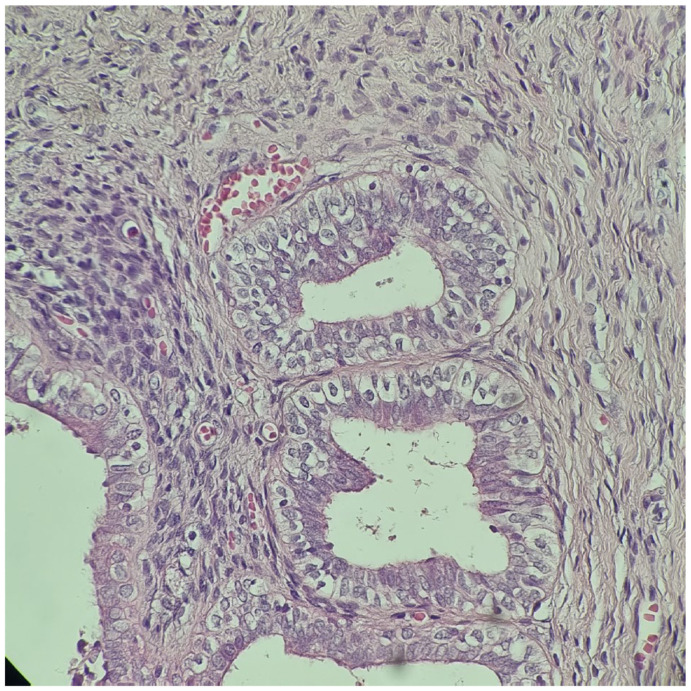
Rudimentary uterus with hypoplastic endometrial and myometrial layers. H&E stain, 400×.

**Figure 3 diagnostics-14-02621-f003:**
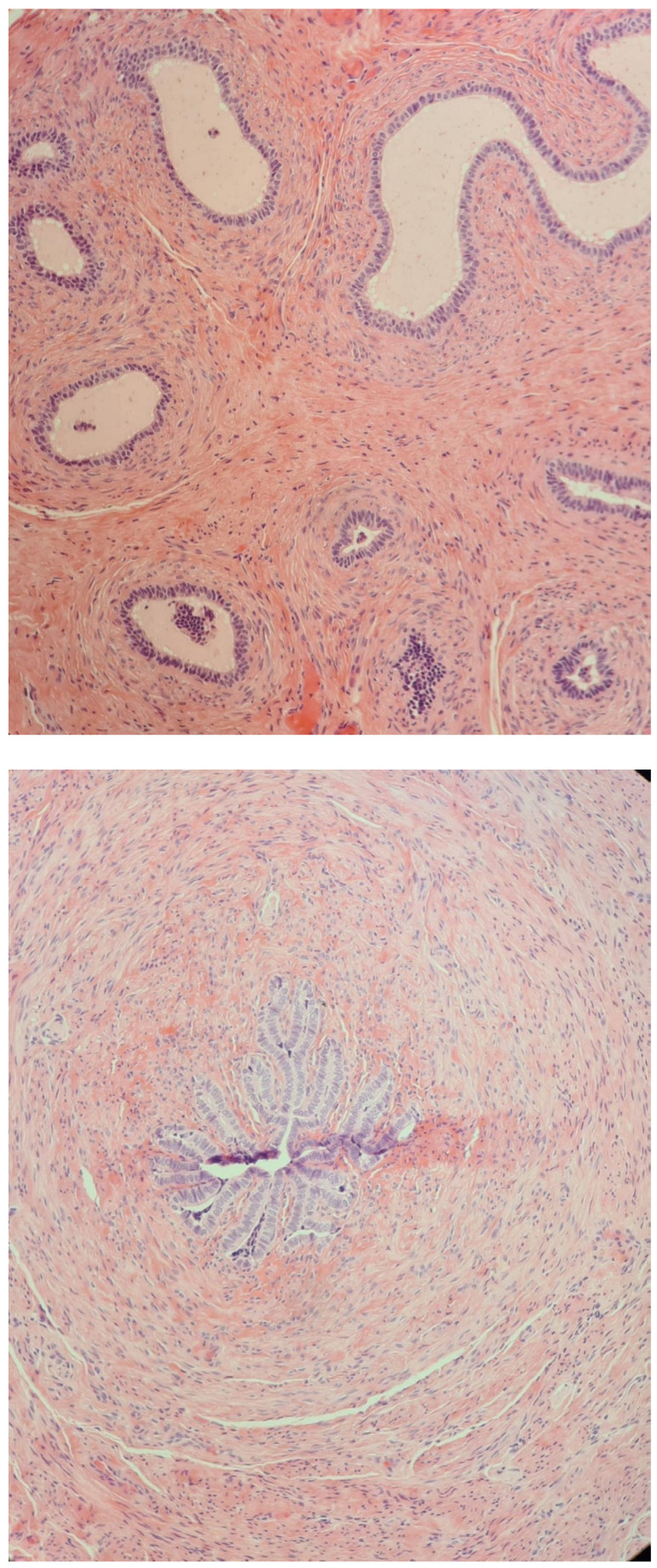
Structure of the hypoplastic vas deferens and epididymis with dilated tubules (2021), H&E stain, 200× and 400×.

**Figure 4 diagnostics-14-02621-f004:**
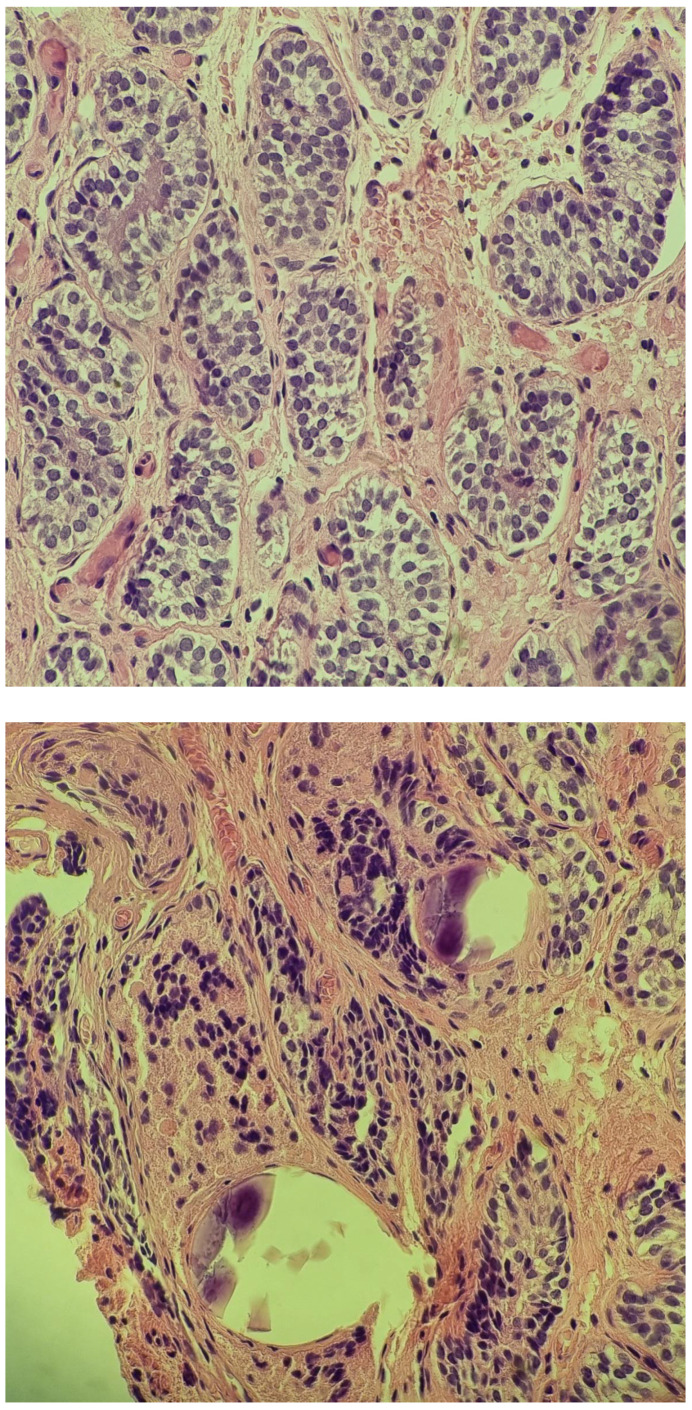
Tumor in the inguinal region: Immature testis consisting of small seminiferous tubules composed of mature and immature Sertoli cells, with no evidence of spermatogenesis; scattered microliths. H&E stain, 100× and 200×.

**Figure 5 diagnostics-14-02621-f005:**
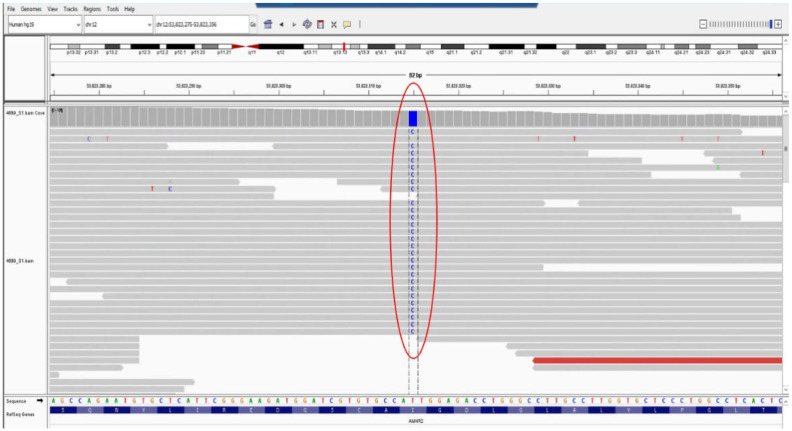
Integrative Genomics Viewer (IGV) showing the homozygous variant NC_000012.11:g.53823315T>C in exon 8 of the *AMHR2* gene in the patient.

**Figure 6 diagnostics-14-02621-f006:**
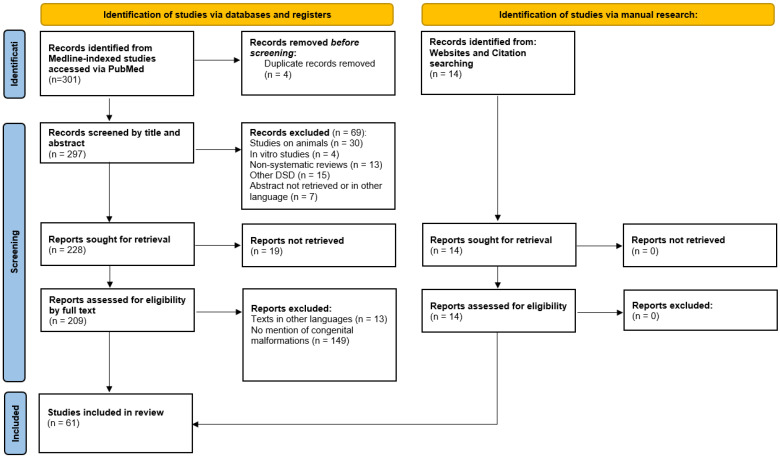
Flowchart depicting the search strategy employed to find the studies included in the review (following 2020 PRISMA guidelines).

**Table 1 diagnostics-14-02621-t001:** Malformations associated with PMDS reported in the literature (from PubMed-indexed articles, accessed on 1 February 2024).

Malformations Affecting	Cases	Genetic Testing (If Available)	Other Associated Malformations	Observations	References
Male excretory ducts					
Vas deferens/epididymis dissociation, agenesia, fusion, duplication, etc.	29			Not specific to PMDS, often associated with cryptorchidism	[6,7,8,9,10,11,12,13,14,15,16,17,18,19,20,21,22]
**Testes**					
Fused testes	2			There is also a case of demonstrated fusion of the testes after birth	[14,23]
Testicular regression syndrome (TRS)	9		Late regression without hypospadias	We excluded cases with a history of testicular torsion, inguinal hernia incarceration, or surgery in the inguinal–scrotal region	[20,24,25,26,27,28,29,30,31,32,33,34,35,36]
Polyorchidism	3	AMHR2 mutation in the index case. No genetic testing in other cases.			[2,3]
**Urinary system**					
Crossed fused renal ectopia	1				[37]
Multicystic kidney	1				[38]
Kidney ectasia	1				[39]
Kidney hypoplasia	1		Intestinal lymphangiectasia, prenatal growth deficiency, hypertrophied alveolar ridges, redundant nuchal skin, and hepatomegaly		[40]
Hydronephrosis	3		Intestinal lymphangiectasia (2), pulmonary lymphangiectasia (1), prenatal growth deficiency (2), hypertrophied alveolar ridges (2), redundant nuchal skin (1), postaxial polydactyly (1), and hepatomegaly (2)		[40,41]
Vesico-uretral reflux	1				[42]
Ureteral duplication	1				[43]
**Gastrointestinal tract**					
Intestinal lymphangiectasia/lymphangiomyxoma	8		Prenatal growth deficiency (3), hypertrophied alveolar ridges (3), redundant nuchal skin (4), postaxial polydactyly (2), renal anomalies (2—hydronephrosis and 1—renal hypoplasia), and hepatomegaly (4)		[40,44,45,46]
			Pulmonary lymphangiectasia (2)		
Hirschsprung disease	2				[47,48]
Esophageal atresia	1	Idiopathic PMDS (PPP1R12A)			[49]
Jejunal atresia	1				[50]
Ileal atresia	3	Idiopathic PMDS (PPP1R12A)			[49]
**Spleen**					
Polysplenia	1		Short pancreas (no body or tail) and unilateral TRS		[24]
**Vascular system**					
Pulmonary artery stenosis	1		Facial dysmorphism		[51]
Mesenteric vein abnormality	1	Trisomy 7	Glaucoma		[52]
**Nervous system**					
Cerebellar ataxia and optic nerve atrophy	1				[52]
Mental retardation	1	Monosomy 11	Glaucoma and aniridia		[52]
**Sex chromosome abnormalities**					
Klinefelter syndrome	3	47XXY (2), 46XY/47XXY (1)			[53,54,55]
**Adrenal glands**					
Congenital adrenal hyperplasia	2		17 alpha-Hydroxylase deficiency (1) and 21 alpha-Hydroxylase deficiency (1)		[25,56]
**Metabolic abnormalities**					
Lipoatrophic diabetes and vitamin D-resistant rickets	1	Normal AMH gene			[57]

## Data Availability

The data presented in this study are available from the corresponding author upon request. The data are not publicly available due to the confidentiality of personal data.

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
