# Peer review of "Persistent Müllerian Duct Syndrome with Supernumerary Testicles Due to a Novel Homozygous Variant in the AMHR2 Gene and Literature Review"

_diagnostics, 2024, doi:10.3390/diagnostics14232621_

Round 1
Reviewer 1 Report
Comments and Suggestions for Authors
Very intersting paper relating to an uncommon topic which can present itself under clinical observation. The PMDS and supernumerary testicles coexistence, which must push the clinician to further investigate, has been very well highlighted, as well as the correlation with the homozygous pathogenic variant in AMHR2 gene, the first described to date in Literature
Author Response
Dear Reviewer,
Thank you for your comments.
Reviewer 2 Report
Comments and Suggestions for Authors
The manuscript “Persistent Müllerian duct syndrome with supernumerary testicles due to a novel homozygous variant in AMHR2 gene and literature review” presents an interesting Case Report and an extensive literature review. Its publication would be interesting for the medical community, endocrinology clinicians, urologists and geneticists.
However, several points need to be corrected:
- 1) The Abstract should include the specific homozygous AMHR2 mutation detected.
- 2) No information is given on parents possible consanguinity, nor possible molecular study.
- 3) References should be given either alphabetical or numerical.
- 4) Give LH, FSH and testosterone levels when AMH, INHB and tumoral markers where performed.
- 5) Anthropometric data should be given at the last evaluation: height, weight, genital evaluation, pubertal stage.
- 6) Fig. 4 is not demonstrative of the mutation, nor the legend included. Give the right figure and legend.
- 7) Description and citation of Fig. 5 and Table 1 should be included in the text, at the end of the Genetic Testing.
- 8) Ref, number 23, Chirita-Emandi 2020 is not cited.
- 9) Ref. Manjunath 2010 and Ratil 2013 cited on p. 13, are not included among the references.
Comments on the Quality of English Language
Minor corrections
Author Response
Comment 1: The Abstract should include the specific homozygous AMHR2 mutation detected.
Dear Reviewer,
Thank you for your comment. We added in the abstract the specific homozygous AMHR2 mutation detected (I marked it in red).
Comment 2: No information is given on parents possible consanguinity, nor possible molecular study.
Thank you for pointing this out. We added in the manuscript the information regarding parents’ consanguinity. Unfortunately, the molecular study couldn’t be performed in his parents due to insufficient funding. We also added this clarification in the genetic testing subchapter.
Comment 3: References should be given either alphabetical or numerical.
Thank you for pointing this out. We modified the bibliography in alphabetical order.
Comment 4: Give LH, FSH and testosterone levels when AMH, INHB and tumoral markers where performed.
We added in the manuscript LH, FSH and testosterone levels with the normal values according to Tanner stage.
Comment 5: Anthropometric data should be given at the last evaluation: height, weight, genital evaluation, pubertal stage.
We added in the manuscript the anthropometric data.
Comment 6: Fig. 4 is not demonstrative of the mutation, nor the legend included. Give the right figure and legend.
We changed figure 4 and modified the legend as requested.
Comment 7: Description and citation of Fig. 5 and Table 1 should be included in the text, at the end of the Genetic Testing.
We inserted the description and citation of figure 5 and table 1 after the genetic testing.
Comment 8: Ref, number 23, Chirita-Emandi 2020 is not cited.
We inserted the reference in the bibliography.
Comment 9: Ref. Manjunath 2010 and Ratil 2013 cited on p. 13, are not included among the references.
I wasn’t able to find the references cited in the text.
I attached the revised manuscript below.

Reviewer 3 Report
Comments and Suggestions for Authors
Persistent Müllerian duct syndrome (PMDS) is a rare condition in boys with normally virilised external genitalia in whom a uterus and Fallopian tubes are found as a consequence of defects in anti-Müllerian hormone (AMH) secretion by the fetal testis or action in the fetal Müllerian ducts. While an association with testicular regression has been reported as an increased risk, the existence of polyorchidism has been reported only twice in patients with PMDS. The authors report the case of a boy with PMDS in whom more than two testes were apparently found. A review of the literature was performed for the association between PMDS and other congenital anomalies.
Major issues:
The report of the case is not sufficiently informative to support the claim that the patient had more than two testes. The whole report needs to be improved:
1. At the age of 4 years:
a. Which was the presumptive diagnosis at the age of 4 years when the patient was operated on for bilateral cryptorchidism and right orchiopexy and herniorraphy were performed? Were the uterus and Fallopian tubes not seen? This is unusual in PMDS, where the uterus and Fallopian tubes are usually discovered at orchiopexy, given the tight association of the testes with the Fallopian tubes.
b. Why was only right (and not left) orchiopexy performed?
c. Were any hormonal test to assess gonadal function performed at that moment?
2. At 5 years:
a. Explain the reasoning to explain why the right testis ascended again, and explain the surgical techniques applied to solve recurrent cryptorchidism.
b. Explain the presumptive diagnosis when the medially located structure resembling a uterus was found, and the rationale for performing a FISH for SRY.
c. Give technical details regarding the surgical removal of the uterus and Fallopian tubes. This is especially important given the risk of damaging the Wolffian derivatives and/or the testis blood supply.
d. Give further details, with precise chronology, regarding left and right testis position after each surgical procedure.
e. Were any hormonal laboratory tests performed to assess testicular function before and after surgery, in order to drive the aetiological diagnosis?
3. At 9-12 years:
a. Explain clearly where the testes were, with precise chronology. Stating that recurrent left cryptorchidism required reinterventions in 2018 and 2021 is too vague for the reader to critically assess the condition and follow-up.
b. The fact that the structure removed in 2021 was a left testicular remnant is not adequately supported by evidence. This is essential to claim that the patient had more than two testes.
4. At 13 years:
a. Explain why was DSD suspected only now.
b. Describe the clinical status of the patient: pubertal maturation according to Tanner staging.
c. Serum levels of all hormones should be reported, with their reference ranges for Tanner stage rather than for age.
5. Histologic images: they are all insufficient. They need to be provided with sufficient enlargement for the reader to see the histologic and cellular features. They should be processed by a specialist in histology and microscope imaging. Providing histologic images of the structure removed in 2021 and claimed to be testicular tissue is critical to support the conclusion of this work.
6. Genomic analysis is also critical to prove the aetiologic diagnosis. Therefore, the sequencing analysis, bioinformatics filtering strategy, data interpretation and reporting cannot be presented elsewhere, but should be part of the present manuscript.
7. Literature review:
a. It seems to be a systematic review according to Figure 5. However, no comment is made on the results in the “Results” section of the manuscript.
b. The search does not seem adequate (Table 1): many patients with hypospadias were retrieved. Hypospadias and other signs of undervirilisation are clear evidence to rule out PMDS in patients with Müllerian remnants (DSD due to testicular dysgenesis is the most probable diagnosis).
8. The Conclusion is not in line with the objective and does not emerge from the Results. Conversely, it is rather a conclusion of the review of the literature, which is not adequate for a case report. According to CARE guideline, that the authors are recommended to follow, case reports should leave a take-home message or lesson learned from the case. Lessons learned from the literature are adequate for reviews.
Comments on the Quality of English Language
English style shoulñd be proofread by a native speaker with medical skills or by a professional editor.
Author Response
Dear Reviewer,
Thank you for all your comments. We followed your suggestions and provided a comprehensive timeline of diagnostic and surgical interventions, along with detailed clinical and histological findings.
Major issues:
The report of the case is not sufficiently informative to support the claim that the patient had more than two testes. The whole report needs to be improved:
- At the age of 4 years:
- Which was the presumptive diagnosis at the age of 4 years when the patient was operated on for bilateral cryptorchidism and right orchiopexy and herniorraphy were performed? Were the uterus and Fallopian tubes not seen? This is unusual in PMDS, where the uterus and Fallopian tubes are usually discovered at orchiopexy, given the tight association of the testes with the Fallopian tubes.
In February 2013, at the age of 4, the patient was admitted to another clinic, where he was investigated and operated on. From the hospital discharge medical letter, it appears that the diagnosis was "Right cryptorchidism, left testicular agenesis, bilateral inguinal hernia". At that time, the patient had bilateral inguinotomy under general anesthesia. Right orchidopexy with pelvic fixation, exploration of the left inguinal region and bilateral herniorrhaphy were performed. During the exploration of the left inguinal region, the left testicle was not detected, so that the left orchidopexy could not be performed at that time.
- Why was only right (and not left) orchiopexy performed?
During the exploration of the left inguinal region, the left testicle was not detected, so that the left orchidopexy could not be performed at that time.
- Were any hormonal test to assess gonadal function performed at that moment?
In April 2013, in the same clinic where the patient was initially operated on, TSH, FT4, and prolactin were collected with normal values. No other hormonal tests were performed in order to assess the gonadal function.
- At 5 years:
- Explain the reasoning to explain why the right testis ascended again, and explain the surgical techniques applied to solve recurrent cryptorchidism.
In May 2014, at the age of 5, the patient presented to our clinic ( "Maria Sklodowska Curie” Emergency Hospital for Children, Bucharest, Romania). The preoperative investigation (MRI) revealed the presence of a left paramedian retrovesical lesion, possibly the left ectopic testicle, as well as a right testicle located in the upper 1/3 of the right inguinal canal.
The Pregnyl stimulation test indicated viable testicular tissue.
Under general anesthesia, exploratory laparoscopy was performed, which revealed a left intra-abdominal testicle with a modified appearance and an anatomical structure of the Mullerian uterine remnant type. The right testicle was not detectable in the abdominal cavity. A biopsy of the left ectopic testicle was performed (HP: immature testis consisting of small seminiferous tubules without lumen and composed of immature Sertoli cells, no evidence of spermatogenesis), as well as an iterative right inguinotomy, which detected in the right inguinal canal a testicle fixed at this level (probably during the first surgical intervention, which was not performed in our clinic), and a right orchidopexy was carried out.
AMH is involved in the swelling reaction of the genito-inguinal ligament. In PMDS, this process is altered and the gubernaculum remains elongated, similar to the round ligament. Consequently, testes may reascend after orchidopexy and this explains the necessity of recurrent bilateral orchidopexies in our patient.
- Explain the presumptive diagnosis when the medially located structure resembling a uterus was found, and the rationale for performing a FISH for SRY.
The anatomical structure identified during the exploratory laparoscopy had the macroscopic appearance of a uterine rudiment; his position was also typical for this type of diagnosis. Therefore, the suspicion of a disorder of sexual development was raised.
- c. Give technical details regarding the surgical removal of the uterus and Fallopian tubes. This is especially important given the risk of damaging the Wolffian derivatives and/or the testis blood supply.
In October 2014, the laparoscopic excision of the uterine rudiment structure was performed, as well as the left orchidopexy (of the left testicle with a modified, hypotrophic appearance, but anatomically-pathologically interpreted as an immature, prepubertal testicle).
- d. Give further details, with precise chronology, regarding left and right testis position after each surgical procedure.
In April 2018, at the age of 9, we observed that the left testicle had an elevated position, being at the level of the external opening of the left inguinal canal. Therefore, surgery was repeated through an incision at the level of the left hemiscrotum and the testicle was descended. On this occasion, we observed that the left testicle was hypotrophic. The right testicle was located in the right hemiscrotum.
In September 2021, at the age of 12, surgery was performed again on the left hemiscrotum and a small atrophic left testicle was discovered, for which an orchidectomy was performed, taking into account the risk of malignancy. The right testicle was still in the right scrotal bursa (HP: epididymis, ductus deferens, without specific features of masculine gonad, seminiferous tubules respectively that were initially seen at the biopsy performed in May 2014 from the left undescended testis).
In February 2022, at the age of 13, following a routine ultrasound, a nodular image was discovered in the left inguinal region, but outside the left inguinal canal, approximately 10-15 mm deep from the skin plane, which was interpreted to be a supernumerary left testicle and a biopsy of the lesion was recommended. In March 2022, surgery was performed and the left inguinal region was explored, where OUTSIDE THE INGUINAL CANAL, IN FRONT OF IT, a structure similar to a testicle was detected, from which a biopsy was performed. Histopathological diagnosis of the biopsied lesion: immature testis.
- Were any hormonal laboratory tests performed to assess testicular function before and after surgery, in order to drive the aetiological diagnosis?
The Pregnyl stimulation test was performed to assess the testicular function.
- At 9-12 years:
- Explain clearly where the testes were, with precise chronology. Stating that recurrent left cryptorchidism required reinterventions in 2018 and 2021 is too vague for the reader to critically assess the condition and follow-up.
The requested details are specified in point 2.d.
- The fact that the structure removed in 2021 was a left testicular remnant is not adequately supported by evidence. This is essential to claim that the patient had more than two testes.
The structure removed during the surgical intervention in 2021 had the macroscopic characteristics of an atrophic testicle. It was in the continuation of the vas deferens. From the beginning, this testicle had smaller dimensions compared to the right testicle and a modified aspect, identified during the laparoscopic exploration performed in 2014. In all the other surgical reinterventions performed (2014, 2018, 2021) it was observed macroscopically that the volume of this testicle was smaller and smaller, until its total atrophy.
- At 13 years:
- Explain why was DSD suspected only now.
The suspicion of DSD was raised in 2014, but we didn’t have the means to investigate the etiology at that moment.
- Describe the clinical status of the patient: pubertal maturation according to Tanner staging.
At the first presentation in a pediatric endocrinology compartment (Elias Hospital) at the age of 13 years and 4 months the patient was Tanner 2. The right testis was palpable in the scrotum (5 ml), while the left one was identified in the inguinal area.
- Serum levels of all hormones should be reported, with their reference ranges for Tanner stage rather than for age.
FSH=3.03 mIU/ml (0.4-4.6), LH=0.434 mIU/ml (0.1-7.8), testosterone=3.12 ng/dl (Tanner 2: 2.5-432).
We added them in the manuscript.
- Histologic images: they are all insufficient. They need to be provided with sufficient enlargement for the reader to see the histologic and cellular features. They should be processed by a specialist in histology and microscope imaging. Providing histologic images of the structure removed in 2021 and claimed to be testicular tissue is critical to support the conclusion of this work.
We changed the previous images. They were processed by a specialist.
- Genomic analysis is also critical to prove the aetiologic diagnosis. Therefore, the sequencing analysis, bioinformatics filtering strategy, data interpretation and reporting cannot be presented elsewhere, but should be part of the present manuscript.
We made the requested changes in the manuscript in the genetic testing subchapter. They are written in red.
- Literature review:
- It seems to be a systematic review according to Figure 5. However, no comment is made on the results in the “Results” section of the manuscript.
The results are described in Table 1.
- The search does not seem adequate (Table 1): many patients with hypospadias were retrieved. Hypospadias and other signs of undervirilisation are clear evidence to rule out PMDS in patients with Müllerian remnants (DSD due to testicular dysgenesis is the most probable diagnosis).
Thank you for pointing this out. We also thought about it. The primary aim of our review was to assess the association of PMDS with congenital anomalies, mainly supernumerary testes. Therefore, we used the following key words: ‘Persistent Müllerian duct syndrome’ + ‘congenital anomalies’, ‘supernumerary testes’, ‘polyorchidism’, ‘genetic testing’, ‘AMHR2 gene mutations’.
- The Conclusion is not in line with the objective and does not emerge from the Results. Conversely, it is rather a conclusion of the review of the literature, which is not adequate for a case report. According to CARE guideline, that the authors are recommended to follow, case reports should leave a take-home message or lesson learned from the case. Lessons learned from the literature are adequate for reviews.
Thank you for your suggestion. Our first statement in the conclusion that high degree of suspicion and awareness is needed to diagnose this condition refers to the fact that our patient underwent iterative surgeries until a definitive diagnosis was achieved.
We excluded the second phrase regarding fertility and the risk of malignancy.
I attached below the revised manuscript.

Round 2
Reviewer 3 Report
Comments and Suggestions for Authors
The authors have significantly imporved thr quality of the case report.
There is, however, a major concern about the conclusion on the existence of 3 testes. At the age of 12 years, the tissue that was removed from the left side did not have gonadal tissue. Can the authors provide supporting evidence that the left testicular tissue found at the age of 13 at the inguinal level is not the only left testis, rather than a second left testis? Is it not possible that there was a testicular-epidymal dissociation on the left side, with the epidydimis and deferens being removed at 12 years, and the testis at 13 years?
The concerns on the literature review were inadequately addressed. Please see my comments to version 1.
Author Response
Revision letter
Comment 1: The authors have significantly improved the quality of the case report.
There is, however, a major concern about the conclusion on the existence of 3 testes. At the age of 12 years, the tissue that was removed from the left side did not have gonadal tissue. Can the authors provide supporting evidence that the left testicular tissue found at the age of 13 at the inguinal level is not the only left testis, rather than a second left testis? Is it not possible that there was a testicular-epidymal dissociation on the left side, with the epidydimis and deferens being removed at 12 years, and the testis at 13 years?
Dear Reviewer,
Thank you for your comments. In case of a testicular-epididymal dissociation (on the left side), the testicle remains fixed into the scrotum, while the vas deferens and the epididymis ascends into the inguinal canal. In our case, the epididymis was found into the scrotum (at the age of 12) and the testicle was found outside (antero-lateral to) the inguinal canal (at the age of 13). Apart from this, the dimension of the left testicle (that was initially intraabdominal and was descended and fixed into the scrotum at the age of 5) was smaller than the right testicle, was only a few mm and was decreasing in evolution (observed at the age of 9), until it was crumbled (at the age of 12). The testicle found outside the left inguinal canal at the age of 13 was bigger than 1.5 cm, so it cannot be the same as the testicle descended and fixed into the left scrotum.
Comment 2: The concerns on the literature review were inadequately addressed. Please see my comments to version 1.
7a. It seems to be a systematic review according to Figure 5. However, no comment is made on the results in the “Results” section of the manuscript.
We followed your suggestions and provided a description of the results.
- The search does not seem adequate (Table 1): many patients with hypospadias were retrieved. Hypospadias and other signs of undervirilisation are clear evidence to rule out PMDS in patients with Müllerian remnants (DSD due to testicular dysgenesis is the most probable diagnosis).
Dear Reviewer,
We agree with you that hypospadias and other signs of undervirilisation are clear evidence to rule out PMDS in patients with Müllerian remnants (DSD due to testicular dysgenesis is the most probable diagnosis). We mentioned in the discussion part in the manuscript that “The coexistence of Mullerian and Wolffian duct remnants can be encountered in other forms of DSD apart from PMDS. Such examples are partial testicular dysgenesis or ovotesticular DSD (Ahmed et al., 2021; Cools et al., 2018; León et al., 2019; Wherrett, 2015). In contrast to PMDS, these conditions generally present with ambiguous genitalia and asymmetrically formed internal genital organs (Josso & Rey, 2020). …In our case, complete external virilization pointed towards PMDS”.
Nevertheless, our search identified 3 patients with AMHR2 mutation and hypospadias. Therefore, we decided to include the reports.
Round 3
Reviewer 3 Report
Comments and Suggestions for Authors
Unfortunately, the explanation on the existence of a 3rd testis is not completely convincing, and it is not mentioned in the text of the manuscript. Every evidence that the authors consider important should be in the mansucript, and not only in the explanation to the reviewer.
The review of the litearture is inadequately reported. If the authors agree that PMDS cannot occur when there are other signs of undervirilisation (which indicate testicular dysgenesis rather than PMDS), they should mention that in the text and eliminate the references retrieved that incorrectly classify patients as PMDS. Otherwise, the reader understands that those patients have PMDS.
Comments on the Quality of English LanguageEnglish style is acceptable.
Author Response
Revision letter
Comment 1: Unfortunately, the explanation on the existence of a 3rd testis is not completely convincing, and it is not mentioned in the text of the manuscript. Every evidence that the authors consider important should be in the mansucript, and not only in the explanation to the reviewer.
Dear Reviewer,
Thank you for your comments. We think that the testes found outside the inguinal canal is a supernumerary testis and we added the explanation in the manuscript. The left testis that was initially intraabdominal and was descended and fixed into the scrotum at the age of 5 decreased in size in evolution (as it was discovered at the age of 9 when an iterative orchidopexy was performed) and finally crumbled (at the age of 12 the tissue that was removed had no gonadal tissue).
Comment 2: The review of the litearture is inadequately reported. If the authors agree that PMDS cannot occur when there are other signs of undervirilisation (which indicate testicular dysgenesis rather than PMDS), they should mention that in the text and eliminate the references retrieved that incorrectly classify patients as PMDS. Otherwise, the reader understands that those patients have PMDS.
Dear Reviewer,
We followed your suggestion and removed the references that referred to the association between PMDS and hypospadias considering the fact that undervirilization is an indicator of testicular dysgenesis rather than PMDS.
Round 4
Reviewer 3 Report
Comments and Suggestions for Authors
The bibliography review was corrected as suggested.
Author Response
Comment: The bibliography review was corrected as suggested.
Dear Reviewer,
Thank you for all your comments and suggestions that helped us improve the quality of the manuscript